# DialogQAE: N-to-N Question Answer Pair Extraction from Customer Service Chatlog

**Xin Zheng**[2 4*]    **Tianyu Liu**[5*†]    **Haoran Meng**[1*]    **Xu Wang**[3]    **Yufan Jiang**[3]
**Mengliang Rao**[3]    **Binghuai Lin**[3]    **Yunbo Cao**[3]    **Zhifang Sui**[1†]

[1] National Key Laboratory for Multimedia Information Processing, Peking University
[2] Institute of Software, Chinese Academy of Sciences, China  [3] Tencent Cloud AI
[4] University of Chinese Academy of Sciences, China  [5] Alibaba Group
zhengxin2020@iscas.ac.cn  haoran@stu.pku.edu.cn
{tianyu0421, szf}@pku.edu.cn

## Abstract

Harvesting question-answer (QA) pairs from customer service chatlog in the wild is an efficient way to enrich the knowledge base for customer service chatbots in the cold start or continuous integration scenarios. Prior work attempts to obtain 1-to-1 QA pairs from a growing customer service chatlog, which fails to integrate the incomplete utterances from the dialog context for composite QA retrieval. In this paper, we propose N-to-N QA extraction task in which the derived questions and corresponding answers might be separated across different utterances. We introduce a suite of generative/discriminative tagging based methods with end-to-end and two-stage variants that perform well on 5 customer service datasets and for the first time setup a benchmark for N-to-N DialogQAE with utterance and session level evaluation metrics. With a deep dive into extracted QA pairs, we find that the relations between and inside the QA pairs can be indicators to analyze the dialogue structure, e.g. information seeking, clarification, barge-in and elaboration. We also show that the proposed models can adapt to different domains and languages, and reduce the labor cost of knowledge accumulation in the real-world product dialogue platform. [*].

## 1 Introduction

The development of natural language processing and conversational intelligence has radically redefined the customer service landscape. The customer service chatbots empowered by knowledge bases or frequently asked questions (FAQs) drastically enhance the efficiency of customer support (e.g. Cui et al., 2017; Ram et al., 2018; Burtsev et al., 2018; Liu et al., 2020; Paikens et al., 2020) In the cold start or continuous integration scenarios, harvesting QA pairs from existing or growing customer service chatlog is an efficient way to enrich knowledge bases. Besides the retrieved QA pairs can be valuable resources to improve dialogue summarization (Lin et al., 2021), gain insights into the prevalent customer concerns and figure out new customer intents or sales trends (Liang et al., 2022), which are of vital importance to business growth.

Prior work on question-answer extraction follows the utterance matching paradigm, e.g., matching the answers to the designated questions in a dialogue session (Jia et al., 2020) in the offline setting or figuring out the best response to the specific user query (Wu et al., 2017; Zhou et al., 2018; Yuan et al., 2019) in the online setting. Within this framework, 1-to-1 QA extraction has been explored by Jia et al. (2020), however, we argue that users might not cover all the details in a single query while interacting with the customer service agents, which means that a certain QA pair might involve multiple utterances in the dialogue session.

In this paper, we extend 1-to-1 QA matching to N-to-N QA extraction, where the challenges are two-fold: 1) cluster-to-cluster QA matching with no prior knowledge of the number of utterances involved in each QA pair and the number of QA pairs in each dialogue session, as the question might be distributed in single or multiple user queries. 2) session-level representation learning with a longer context, as the paired questions and answers might be separated within the dialogue, and the model shall detect multiple same-topic questions and then check if the answer is related to any one of the questions. We propose session-level tagging-based methods to deal with the two challenges. Our method is not only compatible with the N-N QA extraction task setting but also 1-1 and 1-N, which is generic. Switching from matching to tagging, we feed the entire dialogue session into powerful pre-trained models, like BERT (Devlin et al., 2019)

---

[*]Our code and data are available at https://github.com/MrZhengXin/DialogQAE

[*]Equal contribution.
[†]Corresponding authors.

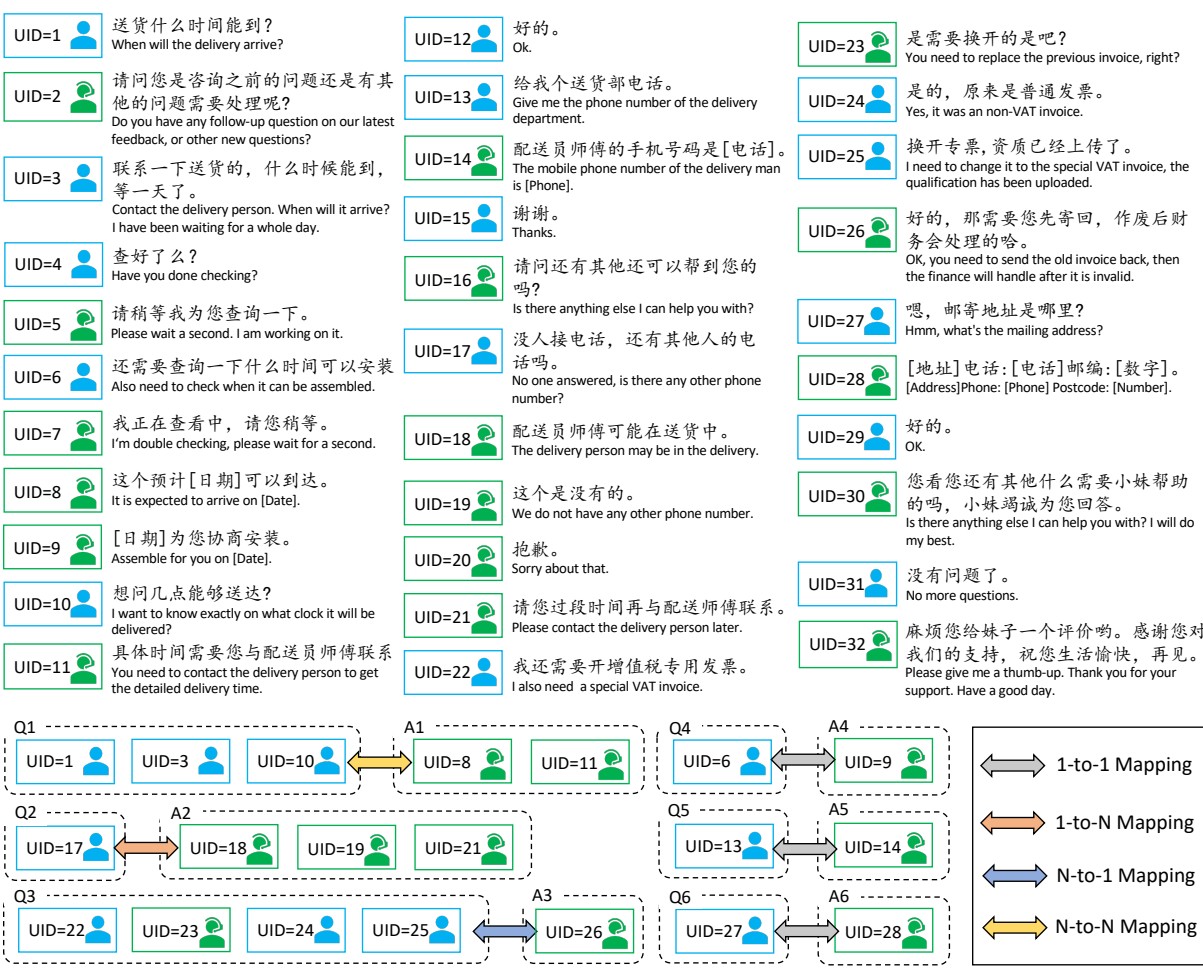

Figure 1: The task overview for DialogQAE. Given a session of two-party conversation with 32 utterances (top), we aim at extracting six QA pairs (bottom) that characterize the dialogue structure and can serve as a valid resource to enrich the knowledge base. The task can be categorized into four types: 1-to-1, 1-to-N, N-to-1, and N-to-N, according to the number of utterances involved in the extracted question or answer unions.

or mT5 (Xue et al., 2021), and design a set of QA tags that empowers N-to-N QA matching. Through careful analysis, we find that DialogQAE can serve as a powerful tool in the dialogue structure analysis, as the relations between and within QA pairs are implicit signals for dialogue actions like information seeking, clarification, barge-in and elaboration. From a pragmatic perspective, we show that the proposed models can be easily adapted to different domains and languages, and largely accelerate the knowledge acquisition on FAQs of real-world users in the product dialogue system. We summarize our contributions below:

- We setup a benchmark for DialogQAE with end-to-end and two-stage baselines that support N-to-N QA extraction, as well as the utterance and session-level evaluation metrics.

- We show that DialogQAE is an effective

paradigm for dialogue analysis by summarizing 5 between-QA-pairs and 3 in-QA-pair relations that characterize the dialogue structure in the conversation flow.

- Through careful analysis of domain and language adaptation, as well as real-world applications, we show that the proposed DialogQAE model effectively automates and accelerates the cold-start or upgrade of a commercial dialogue system.

## 2  Task Overview

A complete snippet of customer service conversation between human service representatives and customers, which is canonically termed as a dialogue session $\mathcal{S}$, consists of multiple, i.e. $n$, dialogue utterances. Formally we have $\mathcal{S} = \{(u_1, r_1), (u_2, r_2), \cdots, (u_n, r_n)\}$, in which $r_i$ sig-

nifies the speaker role of the $i$-th utterance $u_i^{r_i}$. In this paper we focus on two-party dialogue, more concretely we have $r_i \in \{C, A\}$ in which 'C', 'A' represents the roles of speakers: customers and human agents respectively.

After feeding the dialogue session into the DialogQAE model, we expect the model to extract $m$ QA pairs $\mathcal{R} = \{(U_{Q_1}, U_{A_1}), (U_{Q_2}, U_{A_2}), \cdots, (U_{Q_m}, U_{A_m})\}$.

$$U_{Q_j} = \{(u_{q_1}, r_{q_1}), (u_{q_2}, r_{q_2}), \cdots, (u_{q_s}, r_{q_s})\} \tag{1}$$

$$U_{A_j} = \{(u_{a_1}, r_{a_1}), (u_{a_2}, r_{a_2}), \cdots, (u_{a_t}, r_{a_t})\} \tag{2}$$

$U_{Q_j}, U_{A_j}$ represent the unions of question and answer dialogue utterances in $\mathcal{S}$ [2], respectively.

To better characterize the proposed n-to-n dialogue QA extraction, we introduce two notions which are conceptually related to the mapping between dialogue utterances and extracted QA pairs. 1) **Exclusive dialogue utterance**: each utterance in $\mathcal{S}$ can only be exclusively mapped to one single question or answer union in $\mathcal{R}$, i.e. the mapping between $\mathcal{S}$ and $\mathcal{R}$ is a one-to-one (injection) function. 2) **Speaker role consistency**: a common assumption for most two-party conversations is that the customers raise questions while the agents answer the questions. Formally for each extracted QA pair, e.g. $U_{Q_j}$ and $U_{A_j}$ in $\mathcal{R}$, $\{r_{q_{1:s}}\} = \{C\}$, $\{r_{a_{1:t}}\} = \{A\}$. In our setting, the rule of *exclusive dialogue utterance* strictly holds for all the datasets we used. However, although most datasets in this paper exhibit *speaker role consistency*, we still observe the customer queries in the answer union or the agent responses in the question unions, for example in Fig 1 the 23-rd utterance from the agent is included in the question union Q3.

As shown in Fig 1, depending on the sizes of question and answer unions, e.g., $U_{Q_j}, U_{A_j}$, in the certain QA pair, we categorize the DialogQAE task into four types: 1-to-1, 1-to-N, N-to-1 and N-to-N, in which the former and latter numbers indicate the size of question and answer unions respectively.

## 3 Methodology

### 3.1 Tagging as Extraction

The prior mainstream research on dialogue QA matching are based on the text segment alignment, such as matching the specific answers with the

given questions in QA extraction (Jia et al., 2020), measuring the similarity of the user query and candidate answers in the response selection (Wu et al., 2017; Henderson et al., 2019), or extracting relations with entity matching in the dialogue (Tigunova et al., 2021).

The key to successfully excavating n-to-n QA pairs from customer service chatlog is to figure out the cluster-to-cluster mapping among utterances in a dialogue session.

### 3.2 End-to-end QA Extraction

We convert QA extraction into "fill-in-the-blank" sequence labeling task, hoping that the model would quickly learn to predict the label $l_i \in \{O, Q_j, A_j\}$ based on the corresponding utterance $U_i$. After label prediction, we collect the QA pairs $\mathcal{R} = \{(U_{Q_j}, U_{A_j})|U_{Q_j} = \{u_k|l_k = Q_j, 1 \leq k \leq n\}, U_{A_j} = \{u_k|l_k = A_j, 1 \leq k \leq n\}, 1 \leq j \leq m\}$ from the labels $L = \{l_i|1 \leq i \leq n\}$.

As depicted in Figure 2, the input of the model is "$r_1$: $(u_1, r_1)$ [MASK] [SEP] ...$r_n$: $u_n^{r_n}$ [MASK] [SEP] ", where [MASK], [SEP] signifies mask token, separation token and both of which are in the vocabulary of the masked language model. We formulate the QA sequence labeling task as either a generative, i.e. mT5-style (Xue et al., 2021), or a discriminative, i.e. BERT-style (Devlin et al., 2019)), classifier.

From the generative perspective, i.e., span-corruption model mT5, the [MASK] token symbolizes the <extra_id_i> for each utterance $u_i$, and we use the semicolon (;) as the replacement for the separation token [SEP]. The output of the QA extraction model is a list of Q/A labels $L$, where for the encoder-only model each prediction is exactly on the masked position, and for encoder-decoder model mT5 the prediction is a sequence "<extra_id_0> $l_1$... <extra_id_n − 1> $l_n$". For the discriminative tagging model (BERT-style), we use [unusedX] to denote the label set $\{O, Q_1, ..., O_1, ...\}$ and for span-corruption encoder-decoder model, we just use their text form "O, Q1, ..., O1, ..." to represent the label.

### 3.3 Two-stage QA Extraction

Instead of predicting the label of utterance in a single round, we could decouple the process into two steps (Moryossef et al., 2019; Liu et al., 2019, 2021), which firstly figures out questions then extract corresponding answers. We illustrate the two-stage workflow in Figure 6: in the first stage, the

---

[2]In equation 1 and 2, the upper indexes, i.e. $j$, of $q_{1:s}$ and $a_{1:t}$ are omitted for simplicity.

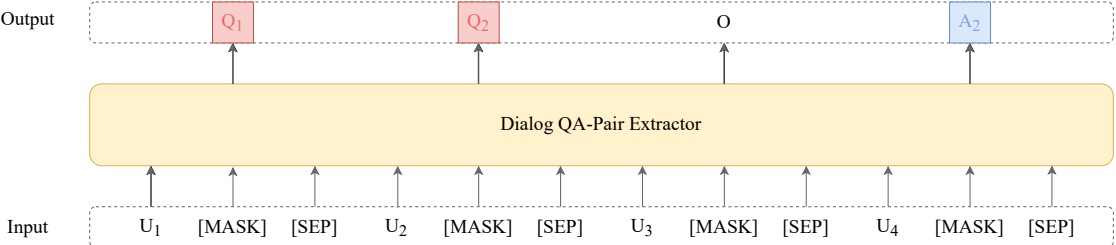

Figure 2: The model workflow for the 'fill-in-the-blank' style one-stage dialog QA pair extraction in the end-to-end fashion. $U_{1:4}$, $Q_{1:2}$, $A_2$ and O represent dialogue utterances, questions, answers and not-Q-or-A utterances.

model input is the same as the end-to-end QA extraction (Sec 3.2), however the dialogue question extractor only predicts $l_{stage1} \in \{O, Q_1, ...\}$, to determine whether each utterance in the dialogue session is a question or not. In the second stage, we fill in the labels where the stage-1 model predicts as questions. Then we feed the filled utterances sequence to the dialog answer extractor, which predicts the remaining utterances within the label set $l_{stage2} \in \{O, A_1, ...\}$, to decide whether they are the answer $A_j$ to the question $Q_j$.

Moreover, in the 1-to-N (including 1-to-1) scenario, where a question covers only a single utterance, we could further break down the question labeling process in a context-less way. As shown in Figure 7, at stage 1, we separately perform binary classification $\{Q, O\}$ for each utterance with the input format of " [CLS] $U_i$", where [CLS] is the classification token, and at stage 2, we relabel those predicted as $Q$ in the sequential order $Q_1, Q_2, ...$ to fill in the blank, and then apply the same dialog answer extracting strategy.

## 4 Experiments

### 4.1 Datasets

We conduct the experiments on 5 Chinese customer service multi-turn dialogue datasets, namely CSDS (Lin et al., 2021), MedQA (Jia et al., 2020), EduQA, CarsaleQA and ExpressQA. CSDS and MedQA are two public dialogue chatlog datasets while the latter three datasets are internal datasets which are accumulated through genuine customer-agent interactions in a commercial industrial dialogue platform. CSDS is derived from JDDC (Chen et al., 2020a) corpus and tailored for dialogue summarization where n-to-n QAs are also provided as the clues for the summaries. MedQA is accumulated on a medical QA platform[3] that covers con-

---
[3] https://www.120ask.com/

versations between doctors and patients. EduQA, CarsaleQA and ExpressQA, as indicated by their names, come from real-world conversations in the education, carsales and express delivery domains. As shown in Table 1, EduQA, CarsaleQA and ExpressQA are composed exclusively of 1-1 QA pairs while CSDS and MedQA involve 1-N, N-1 and N-N mappings in the extracted QA pairs.

### 4.2 Evaluation Metrics

To evaluate the performance at the utterance level, we apply the traditional precision (P), recall (R) and F1 metrics, which ignore the non-QA label "$O$":

$$P = \frac{\sum_i \sum_j \mathbb{I}_{Pred_j^{(i)}!=O, Pred_j^{(i)}=Ref_j^{(i)}}}{\sum_{1 \leq i \leq N} \sum_{1 \leq j \leq n^{(i)}} \mathbb{I}_{Pred_j^{(i)}!=O}},$$

$$R = \frac{\sum_i \sum_j \mathbb{I}_{Ref_j^{(i)}!=O, Pred_j^{(i)}=Ref_j^{(i)}}}{\sum_{1 \leq i \leq N} \sum_{1 \leq j \leq n^{(i)}} \mathbb{I}_{Ref_j^{(i)}!=O}}, \quad (3)$$

$$F1 = \frac{2 * P * R}{P + R},$$

where $N$ is the number of instances, and $Pred^{(i)}$, $Ref^{(i)}$ denote the prediction and reference label sequences of the $i$-th instance.

Similarly, for QA-pair level evaluation, we propose adoption rate (AR), hit rate (HR) and session F1 (S-F1):

$$AR = \frac{\sum_i \sum_j |R\_pred^{(i)} \cap R\_ref^{(i)}|}{\sum_{1 \leq i \leq N} |R\_pred^{(i)}|},$$

$$HR = \frac{\sum_i \sum_j |R\_pred^{(i)} \cap R\_ref^{(i)}|}{\sum_{1 \leq i \leq N} |R\_ref^{(i)}|}, \quad (4)$$

$$S\text{-}F1 = \frac{2 * HR * AR}{HR + AR},$$

where $R\_pred^{(i)}$, $R\_ref^{(i)}$ denote the prediction and reference QA-pair set of the $i$-th instance.

| Dataset | #Sess | Avg_Us | Avg_Qs | Avg_As | Dist_QA | Ratio of QA Pairs(%) | | | |
|---------|-------|--------|--------|--------|---------|------|------|------|------|
| | | | | | | 1-1 | 1-N | N-1 | N-N |
| CSDS | 9100 | 25.99 | 2.26 | 2.52 | 6.88 | 34.43 | 22.58 | 15.61 | 27.41 |
| MedQA | 700 | 36.46 | 9.73 | 10.78 | 2.19 | 70.95 | 29.05 | 0 | 0 |
| EduQA | 3000 | 10.63 | 1.29 | 1.27 | 2.23 | 100 | 0 | 0 | 0 |
| CarsaleQA | 3172 | 10.71 | 0.21 | 0.14 | 1.36 | 100 | 0 | 0 | 0 |
| ExpressQA | 5000 | 14.13 | 0.57 | 0.41 | 2.57 | 100 | 0 | 0 | 0 |

Table 1: Dataset statistics. We list the number of sessions (#Sess), the average number of utterances (Avg_Us), questions (Avg_Qs) and answers (Avg_As) in each session. We also figure out the average distances between the starting and ending utterance within a QA pair (Dist_QAs), which signifies the (minimal) context required for successful QA extraction, as well as the ratio of 1-1, 1-N, N-1, N-N QA pairs in each datasets.

From the perspective of FAQ database population by extracting QA pairs from the customer service chatlog, the predicted QA pairs would serve as an automated module in the workflow, followed by the human verification. The adoption rate (AR) corresponds to the ratio of "accepted" QA pairs by human judges within the predicted QAs, which is analogous to the utterance-level precision. The hit rate (HR), on the other hand, signifies the proportion of predicted QAs in all annotated QAs within the dialogue session which corresponds to the utterance-level recall.

### 4.3 Experimental Settings

We experiment with a variety of pre-trained models via Hugging Face Transformers (Wolf et al., 2020), which are encoder-decoder model mT5 (Xue et al., 2021) with three different parameter scales, namely T5-base (580M), T5-large (1.2B), T5-xl (3.7B), and encoder-only model including chinese-bert-wwm-ext (110M), chinese-roberta-wwm-ext (110M), chinese-roberta-wwm-ext-large (330M) (Cui et al., 2020), Deberta-Chinese-Large (304M) [4], Erlangshen-MegatronBert (1.3B) [5], as the backbones for the end-to-end and two-stage models. For contextless question classification and question-answer matching, we use chinese-roberta-wwm-ext-large (330M) (Cui et al., 2020). We use the Adam optimizer (Kingma and Ba, 2015) with the learning rate of 3e-5 and train the models for at most 9 epochs on 4 Nvidia A100 GPUs.

---

[4] https://huggingface.co/WENGSYX/Deberta-Chinese-Large

[5] https://huggingface.co/IDEA-CCNL/Erlangshen-MegatronBert-1.3B

## 5 Analysis and Discussions

### 5.1 Baseline Performance

Table 2 and 3 illustrate the utterance-level and session-level performance of QA extraction on the MedQA and CSDS datasets respectively. For both end-to-end and two-stage models, enlarging the model parameters leads to a considerable performance gain, which indicates that the dialogue session encoders with higher capacity are of vital importance for extracting QA pairs. In terms of the comparisons between the end-to-end and two-stage models, we observe that the two-stage models outperform end-to-end models on the MedQA dataset while it is the other way around on the CSDS dataset, which shows that end-to-end methods are more favorable in the N-to-N QA extraction that requires reasoning over longer dialogue context, such as CSDS (6.88 in Dist_QA and 65.57% non-1-to-1 QAs as shown in Fig 1), as presumably complex Q-A mapping exaggerates the error propagation of aligning the potential answers to the given predicted questions in the two-stage models. For the model performance on the N-to-N mapping shown in Fig 3, as we expect, the models get higher scores on 1-to-1 mapping than N-to-N mapping.

We also highlight the comparison between the generative (mT5-style, 'Gen') and the discriminative (BERT-style, 'Tag') models in Table 2 and 3. We observe that with comparable pre-trained model size, i.e. DeBERTa-large (304M) versus mT5-base (580M) and MegatronBERT (1.3B) versus mT5-large (1.5B), generative models perform better on the CSDS dataset while discriminative models win on the MedQA dataset, showing that T5 models might be a promising option on the dialogue analysis with long context (Meng et al., 2022). We believe that model size is an important factor in performance, since intuitively model with more

| Training Strategy | Base Model | Utterance Level(%) | | | Session Level(%) | | |
|---|---|---|---|---|---|---|---|
| | | P | R | F1 | AR | HR | S-F1 |
| End-to-End (Gen) | mT5-base | 79.00 | 86.19 | 82.44 | 48.11 | 50.21 | 49.13 |
| End-to-End (Gen) | mT5-large | 87.90 | 91.69 | 89.75 | 66.77 | 68.99 | 67.86 |
| End-to-End (Gen) | mT5-xl | **92.39** | **93.09** | **92.74** | **75.63** | **77.41** | **76.51** |
| End-to-End (Tag) | BERT-base | 79.85 | 81.58 | 80.70 | 48.85 | 48.10 | 48.47 |
| End-to-End (Tag) | RoBERTa-base | 80.73 | 83.45 | 82.07 | 52.05 | 52.05 | 52.05 |
| End-to-End (Tag) | RoBERTa-large | 88.32 | 89.17 | 88.75 | 65.10 | 65.50 | 65.30 |
| End-to-End (Tag) | DeBERTa-large | 88.52 | 89.91 | 89.21 | 66.55 | 67.51 | 67.02 |
| End-to-End (Tag) | MegatronBERT | 89.82 | 90.73 | 90.27 | 71.16 | 70.79 | 70.97 |
| Two-Stage (G+G) | mT5-base | 82.74 | 89.62 | 86.04 | 56.81 | 58.88 | 57.83 |
| Two-Stage (G+G) | mT5-large | 88.86 | 93.05 | 90.90 | 70.06 | 72.07 | 71.05 |
| Two-Stage (G+G) | mT5-xl | 92.86 | 92.97 | 92.91 | 77.62 | 78.70 | 78.15 |
| Two-Stage (B+G) | mT5-base | 82.24 | 89.64 | 85.78 | 57.83 | 58.21 | 58.02 |
| Two-Stage (B+G) | mT5-large | 88.32 | 92.13 | 90.19 | 70.63 | 71.10 | 70.86 |
| Two-Stage (B+G) | mT5-xl | **92.85** | **91.96** | **92.40** | **77.21** | **77.72** | **77.46** |

Table 2: The benchmark for the QA extraction task on the MedQA dataset. The discriminatively (BERT-style) and generatively (mT5-style) trained end-to-end models are abbreviated as 'Tag' and 'Gen'. The 2 variants (B+G and G+G) of two-stage models differ in the model formulation of the first stage, i.e. binary classifier (Fig 6) versus mT5-style generative (Fig 7) model. We highlight the winner in each training strategy and the best scores with boldface and underlined marks.

| Training Strategy | Base Model | Utterance Level(%) | | | Session Level(%) | | |
|---|---|---|---|---|---|---|---|
| | | P | R | F1 | AR | HR | S-F1 |
| End-to-End (Gen) | mT5-base | 82.54 | 54.67 | 65.77 | 20.74 | 23.64 | 22.10 |
| End-to-End (Gen) | mT5-large | 84.38 | 57.55 | 68.43 | 22.04 | 26.86 | 24.22 |
| End-to-End (Gen) | mT5-xl | 84.61 | 57.15 | 68.22 | 22.98 | 26.53 | 24.63 |
| End-to-End (Tag) | BERT-base | 84.98 | 47.30 | 60.77 | 18.77 | 20.00 | 19.37 |
| End-to-End (Tag) | RoBERTa-base | **86.86** | 44.72 | 59.04 | 18.18 | 18.81 | 18.49 |
| End-to-End (Tag) | RoBERTa-large | 86.63 | 44.91 | 59.15 | 18.30 | 19.49 | 18.88 |
| End-to-End (Tag) | DeBERTa-Large | 83.95 | 45.91 | 59.36 | 19.61 | 19.41 | 19.51 |
| End-to-End (Tag) | MegatronBERT | 84.00 | 51.49 | 63.84 | 19.76 | 20.76 | 20.25 |
| Two-Stage (G+G) | mT5-base | 77.39 | 52.89 | 62.84 | 19.15 | 18.73 | 18.94 |
| Two-Stage (G+G) | mT5-large | 80.45 | **56.17** | **66.15** | 20.32 | 22.71 | 21.45 |
| Two-Stage (G+G) | mT5-xl | **83.77** | 54.02 | 65.68 | **22.46** | **24.32** | **23.35** |

Table 3: The benchmark for the QA extraction task on the CSDS dataset. Note that two-stage (B+G) models (Table 2) are incompatible with CSDS as the binary classifier is tailored for 1-to-1 and 1-to-N extraction (Sec 3.3).

parameters would fit the data better, and yet discriminative models with Masked Language Model pre-training task may not enjoy the same scaling law (Hoffmann et al., 2022) as the generative models do.

## 5.2 Dialogue Structure Analysis

Prior research tried to extract and analyze the structure of a given dialogue session through latent dialogue states (Qiu et al., 2020), discourse parsing (Galitsky and Ilvovsky, 2018) or event extraction (Eisenberg and Sheriff, 2020). However, those methods are specific to the predefined semantic/information schema or ontology, i.e., discourse, dependency, AMR parsing trees (Xu et al., 2021, 2022), dialogue actions or event/entity labels (Liang et al., 2022). Through the analysis of the extracted QA pairs of a dialogue session, we summarize a more general schema to categorize the dialogue structure according to the customer-agent interaction in the dialogue flow.

Fig 4 demonstrates the typical 'between-QA-pairs' relations based on the extracted QA mappings. The most common case is Sequential

QA Flow, where Position(A1) < Position(Q2) and Role(Q1) = Role(Q2); in this case, one complete QA pair is after another. For Follow-up Information Seeking, here Position(A1) < Position(Q2) but Role(Q1) ≠ Role(Q2), indicating the answer leads to a new question. For elaboration/Detailing, Position(Q2) < Position(A1) and Role(Q1) = Role(Q2), which means one person asked two questions in a row, and in turn, the other answered consecutively. In the example, the doctor sequentially answers the consecutive questions raised by the patients, with the second answer elaborating on the first one. For Clarification/Confirmation, Position(Q2) < Position(A1) and Role(Q1) ≠ Role(Q2) and Position(A2) < Position(A1), which implies the first question can not get the answer yet, and more information is needed from the questioner; once provided, the first question can finally be answered correctly. In the example, the doctor asked a clarification question on when the symptoms occurred after the inquiry of the patient instead of answering the inquiry instantly. For Barge-in/Interruption, which is not common, is the case of Position(Q2) < Position(A1) and Role(Q1) = Role(Q2) and Position(A2) < Position(A1), where the second question is answered first. As shown in Fig 3, the QA extraction models perform better on SF, FIS, and BI than CC and ED, presumably the interleaving QA pairs pose a bigger challenge to the dialogue information extraction.

We delve into the relative position of the question and answer utterances within an N-to-N QA pair in Fig 5. Most questions and answers are disjoint within a QA pair while the overlapping questions and answers account for 26.91% in the CSDS dataset. We take a further step to split the overlapping QAs into two circumstances: in-pair Q-A and in-pair Q-A-Q, depending on the role (Q or A) of the last utterance in the QA pair. As illustrated in Fig 3, all three QAE models perform better on the disjoint QA pairs than overlapping ones.

### 5.3 Domain and Language Adaptation

We illustrate the domain and language adaptation of our dialogQAE models in Table 4 and 5 respectively, which highlight the real-world utility of our models.

In Table 4, we observe that mixing the datasets from different domains is a simple but effective way to boost the overall performance. The potential

| Dataset | Carsale | Express | Edu | Avg. |
|---|---|---|---|---|
| Carsales | 70.89 | 40.78 | 60.73 | 57.47 |
| Express | 54.39 | **86.33** | 52.91 | 64.54 |
| Edu | 74.05 | 47.52 | **86.31** | 69.29 |
| All | **80.43** | 83.41 | 85.96 | **83.27** |

Table 4: The domain adaptation of dialogQAE models on EduQA, CarsaleQA and ExpressQA. We report the utterance F1 scores (%) of the End-to-end (Gen, mT5-xl) models.

| Domain | P | AR |
|---|---|---|
| MultiDOGO_airline | 91.30 | 86.40 |
| MultiDOGO_fastfood | 91.04 | 85.05 |
| MultiDOGO_finance | 83.71 | 81.50 |
| MultiDOGO_insurance | 93.17 | 90.00 |
| MultiDOGO_media | 86.75 | 83.02 |
| MultiDOGO_software | 87.00 | 84.91 |

Table 5: The illustration for the language transfer of the End-to-end (Gen, mT5-xl) model trained on MedQA. We abbreviate the precision and adoption rate scores in the utterance and session level as 'P' and 'AR'. The scores correspond to the accuracy of the predicted QA pairs, according to the human judges.

correlation between different domains is the key factor of the model performance on the domain transfer, e.g. the bidirectional transfer between the carsale and the education domains gets higher scores than other domain pairs.

Thanks to the multilingual nature of mT5, the models trained on the Chinese datasets can be easily applied to datasets in other languages, e.g. English. We test the Chinese DialogQAE model on different domains of the MultiDOGO dataset (Peskov et al., 2019). As the MultiDOGO dataset does not have Q-A-pair annotations, we ask the human annotator to decide whether the recognized QA pairs by the MedQA-DialogQAE model are eligible according to the semantics in the dialogue flow. We use majority votes among 3 human judges and the inter-annotator agreement (the Krippendorf's alpha) is 0.89.

## 6 Related Work

### 6.1 QA Extraction

For text-based QA extraction, Rajpurkar et al. (2016) proposed the dataset SQuAD 1.1, in which the 100k+ questions were created by crowdworkers on 536 Wikipedia articles. Subsequently, Du and Cardie (2018) created 1M+ paragraph-level

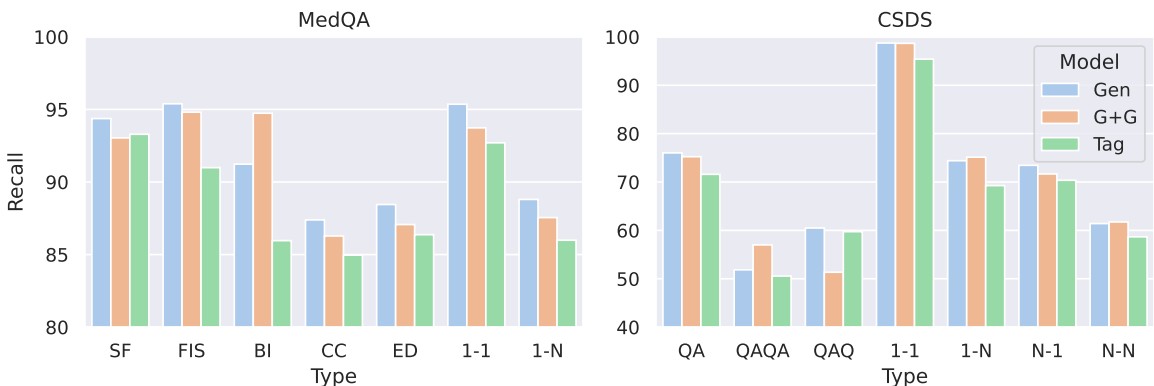

Figure 3: Barplot of recall group by the dialog structure type, on the MedQA and CSDS datasets respectively. Here, we use End-to-End(Gen), mT5-xl model. SF, FIS, BI, CC, ED refer to Sequential QA Flow, Follow-up Information Seeking, Barge-in/Interruption, Clarification/Confirmation, Elaboration/detailing, QA, QAQA, QAQ refer to Disjoint In-pair Q-A, Overlap In-pair Q-A, Overlap In-pair Q-A-Q, and 1-1, 1-N, N-1, N-N refer to 1-to-1, 1-to-N, N-to-1, N-to-N QA matching.

question-answer pairs over 10,000 Wikipedia articles. For question generation, Yang et al. (2017) use a trained model to generate questions on unlabeled data. Later, Wang et al. (2019) proposed to identify key phrases first and then generate questions accordingly. For machine reading comprehension (MRC), research on dialogues MRC aims to teach machines to read dialogue contexts and make response Zeng et al. (2020) aims to answer the question based on a passage as context. Shinoda et al. (2021) leveraged variational question-answer pair generation for better robustness on MRC. However, extraction methods that can work on 1-1, 1-N, and N-N scenario is under-explored.

## 6.2 Dialogue Analysis

For dialogue information extraction (IE), in order to save the efforts of the assessor in the medical insurance industry, Peng et al. (2021) proposed a dialogue IE system to extract keywords and generate insurance reports. To figure out the semantics in the dialogue flow, Galitsky and Ilvovsky (2018) proposed a dialogue structure-building method from the discourse tree of questions. Qiu et al. (2020) incorporated structured attention into a Variational Recurrent Neural Network for dialogue structure induction in an unsupervised way. Eisenberg and Sheriff (2020) introduced a new problem, extracting events from dialogue, annotated the dataset Personal Events in Dialogue Corpus, and trained a support vector machine model.

Relation Extraction over Dialogue is a newly defined task by DialogRE (Yu et al., 2020), which fo-

cuses on extracting relations between speakers and arguments in a dialogue. DialogRE is an English dialogue relation extraction dataset, consisting of 1788 dialogues and 36 relations. MPDD (Chen et al., 2020b) is a Multi-Party Dialogue Dataset built on five Chinese TV series, with both emotion and relation labels on each utterance. Long et al. (2021) proposed a consistent learning and inference method for dialogue relation extraction, which aims to minimize possible contradictions. Fei et al. (2022) introduced a dialogue-level mixed dependency graph. Shi and Huang (2019) proposed a deep sequential model for discourse parsing on multi-party dialogues. The model predicts dependency relations and constructs a discourse structure jointly and alternately.

## 7 Conclusion

In this paper, we propose N-to-N question and answer (QA) pair extraction from customer service dialogue history, where each question or answer may involve more than one dialogue utterance. We introduce a suite of end-to-end and two-stage tagging-based methods that perform well on 5 customer service datasets, as well as utterance and session level evaluation metrics for DialogQAE. With further analysis, we find that the extracted QA pairs characterize the dialogue structure, e.g. information seeking, clarification, barge-in, and elaboration. Extensive experiments show that the proposed models can adapt to different domains and languages and largely accelerate knowledge accumulation in the real-world dialogue platform.

## Limitations

This work focuses on the N-to-N question and answer extraction from a dialogue session and does not touch the relevant tasks such as question generation (e.g. Du et al., 2017; Duan et al., 2017) and dialogue summarization (Lin et al., 2021). The proposed task can be seen as a preparation for the subsequent tasks by decomposing the entire procedure of question generation and dialogue summarization into two steps: extraction before generation. The extracted QA pairs can also be further processed in order to visualize some important factors for customer service, like common customer concerns about the products, winning sales scripts to persuade the customers and emerging or trending user intents (Liang et al., 2022), by a set of atomic natural language processing modules like keyword extraction, sentiment analysis and semantic parsing and clustering.

## Ethics Statement

The internal datasets we used in this paper, i.e. EduQA, CarsalesQA and ExpressQA, have gone through a strict data desensitization process, with the guarantee that no user privacy or any other sensitive data is being exposed by a hybrid of automatic and human verification. Human verification also eliminates the dialogue sessions with gender/ethnic biases or profanities. The other two datasets, CSDS and MedQA, are publicly available and we use them with any modification. The model for extracting questions and answers in the dialogue paves the for N-to-N dialogue QA extraction, without any risk of violating the EMNLP ethics policy.

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

## A  Appendix

### A.1  Model Performance on the Internal Datasets

We show the model performance on the internal datasets, i.e. EduQA, CarsalesQA and ExpressQA in Table 6. In terms of comparison between end-to-end and two-stage models, end-to-end models are clear winner with respect to session level F1 on the Carsales and EndQA datasets, while two-stage models take the lead on the ExpressQA dataset. According to the dataset statistics in Table 1, we guess this is because ExpressQA has longer dialogue session (14.13 for the average number of utterances) and require longer context (2.57 versus 2.23/1.36 in Dist_QA) for extracting QA pairs.

The DialogQAE models have been deployed in a commercial platform for conversational intelligence. The module serves as an automatic dialogue information extractor, followed by human verification and modification on the extracted QA pairs so that they can serve as standard and formal FAQs in the customer service. According to the user feedbacks from the online customer service department of an international express company, the assistance of dialogue QA extraction has largely accelerated the information enrichment for customer service FAQs, reducing from around 8 days per update to 2 days per update.

### A.2  Between-QA-Pairs Relations Examples

We show more examples on the dialogue structure from the MedQA datasets below.

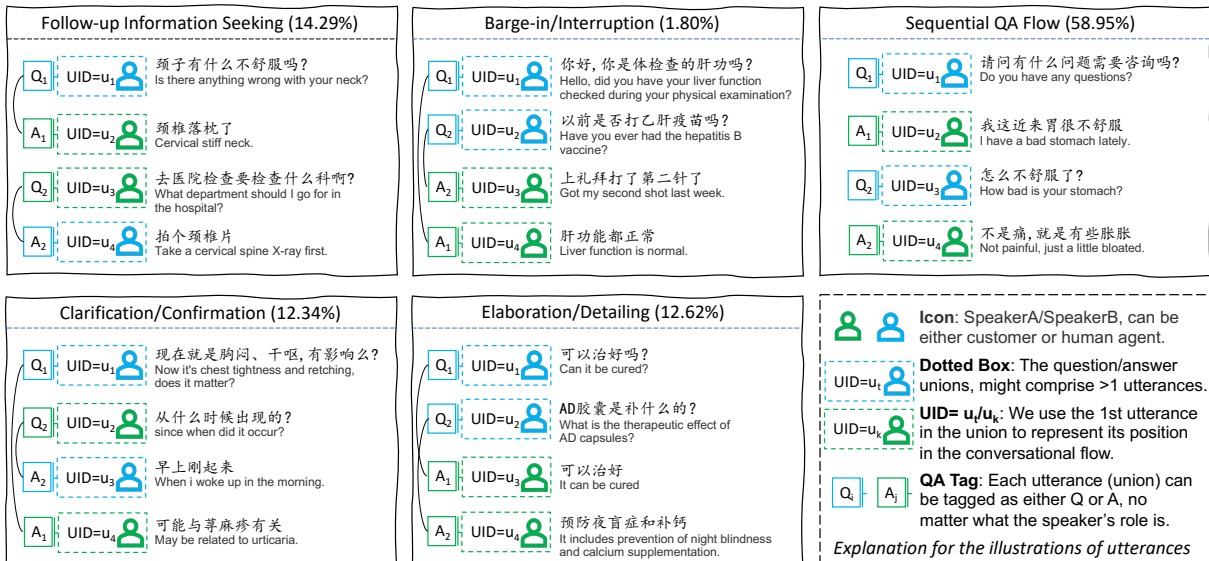

Figure 4: The demonstration for the between-QA-pairs relations and their proportion in the MedQA dataset. Given a snippet of consecutive dialogue utterances (for UIDs, $u_1 < u_2 < u_3 < u_4$), we roughly categorize the dialogue flows into 5 different types according to the between-QA interactions between customers and human agents.

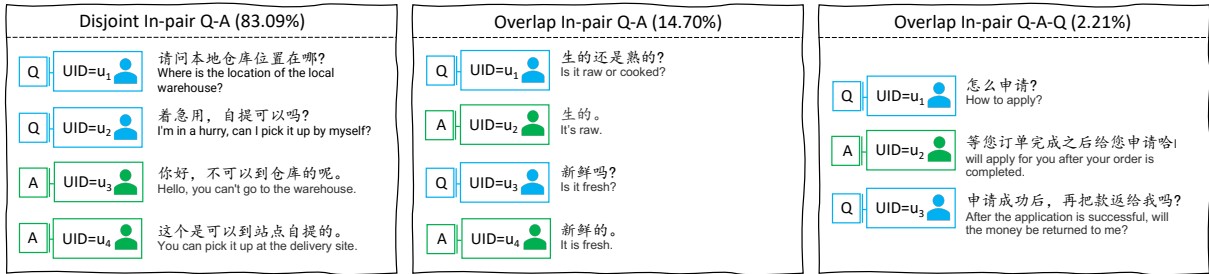

Figure 5: The demonstration of the in-QA-pair relations and their proportion in the CSDS dataset. According to the relative positions of Q and A utterances in an n-to-n QA pair, we categorize the interleaving utterances into 3 types.

| Dataset | Training Strategy | Utterance Level(%) | | | Session Level(%) | | |
|---|---|---|---|---|---|---|---|
| | | P | R | F1 | AR | HR | S-F1 |
| CarsalesQA | End-to-End (Gen) | 83.67 | 61.50 | 70.89 | 42.11 | 44.44 | 43.24 |
| EduQA | End-to-End (Gen) | 88.96 | 83.81 | 86.31 | 55.50 | 65.98 | 60.29 |
| ExpressQA | End-to-End (Gen) | 96.26 | 78.26 | 86.33 | 63.46 | 71.74 | 67.35 |
| CarsalesQA | Two-Stage (G+G) | 85.96 | 73.13 | 79.03 | 43.31 | 53.54 | 47.89 |
| EduQA | Two-Stage (G+G) | 98.44 | 69.23 | 81.29 | 59.62 | 67.39 | 63.27 |
| ExpressQA | Two-Stage (G+G) | 92.99 | 76.54 | 83.97 | 59.66 | 67.31 | 63.25 |

Table 6: The performances of the end-to-end and two-stage models (mT5-large) for the QA extraction task on the internal datasets.

| Role | Utterance |
|---|---|
| Patient | 连续三天头晕晕的,整个人飘飘的,睡一觉就好了,怎么回事啊? |
| | I felt dizzy for three days in a row. My whole body fluttered, but just fine after a night of sleep. What's going on? |
| Doctor | 估计是落枕的原因 |
| | Stiff necks may cause this problem. |
| Doctor | 有过检查吗? |
| | Has there been an inspection? |
| Patient | 没有呢 |
| | No. |
| Doctor | 还有其它症状嘛? |
| | Are there any other symptoms? |
| Patient | 睡不踏实 |
| | I can't sleep well. |
| Patient | 用酒擦可以吗? |
| | Can I wipe myself with alcohol? |
| Doctor | 用温水擦 |
| | Wipe with warm water. |
| Doctor | 痒吗? |
| | Is it itchy? |
| Patient | 不痒 |
| | Not itchy |
| Patient | 怎么治啊? |
| | How to treat it? |
| Doctor | 注意饮食(多吃蔬菜、水果。多喝水。少吃多脂、多糖、辛辣刺激性食物),别熬夜。少化妆。治疗:外涂夫西地酸乳膏(白天涂一次),阿达帕林凝胶(晚上睡前涂一次)。内服丹参酮胶囊 |
| | Pay attention to your diet (Eat more vegetables and fruits. Drink more water. Eat less fatty, sweet and spicy food), and don't stay up late. Wear less makeup. Treatment: external application of fusidic acid cream (once during the day) and adapalene gel (once before bedtime at night). Orally take tanshinone capsules. |

Table 7: Follow-up Information Seeking examples in MedQA

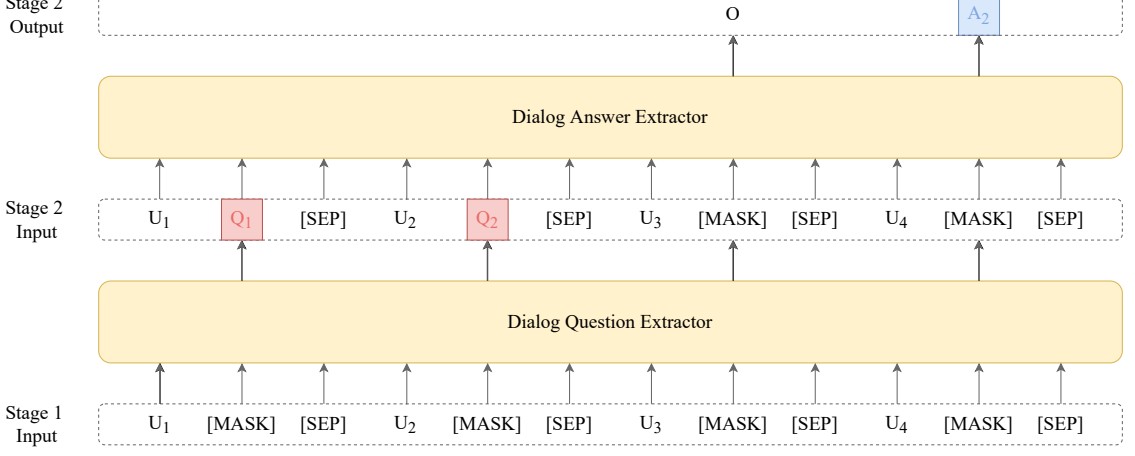

Figure 6: The model workflow for the two-stage QA extraction. The first stage is to extract questions while the second stage corresponds to answer extraction given the predicted questions. Hereby we only illustrate the 'fill-in-the-blank' style question extractor, where the binary question classifier is shown in Fig 7.

| Role | Utterance |
|------|-----------|
| Doctor | 吃过什么药呢? |
| | What medicine have you taken? |
| Doctor | 现在还吃着吗? |
| | Are you still taking them? |
| Patient | 还在吃 |
| | Still taking. |
| Patient | 酚麻美敏片 |
| | Paracetamol, Pseudoephedrine Hydrochloride, Dextromethorphan Hydrobromide and Chlorpheniramine Maleate Tablets |
| Doctor | 多长时间了? |
| | How long has it been like this? |
| Doctor | 多大年龄? |
| | How old are you? |
| Patient | 三十三了 |
| | I'm thirty-three. |
| Patient | 十多年了 |
| | It has been more than ten years. |
| Doctor | 你的宝宝现在有咳嗽吗? |
| | Does your baby have a cough right now? |
| Doctor | 有发热的情况吗? |
| | Does it have a fever? |
| Patient | 没有发热 |
| | No fever |
| Patient | 有咳嗽 |
| | Have a cough. |

Table 8: Barge-in/Interruption examples in MedQA

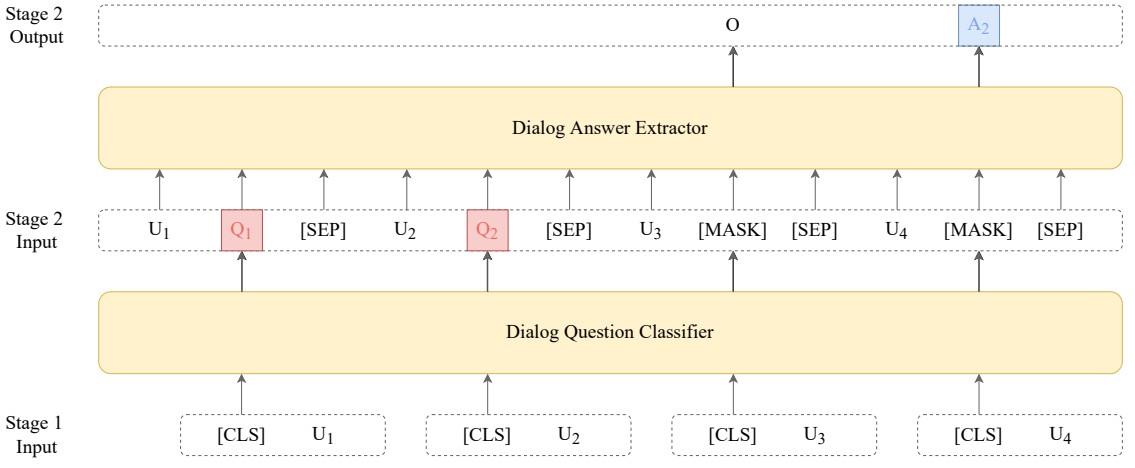

Figure 7: The model workflow for the two-stage QA extraction where the first stage is the binary question classifier. The workflow only works for 1-to-1 or 1-to-N QA extraction.

| Role | Utterance |
|------|-----------|
| Patient | 去医院检查要检查什么科啊? |
| | Which department should I go to the hospital to check? |
| Doctor | 拍个颈椎片 |
| | Take a cervical spine X-ray. |
| Patient | 会不会是睡眠不足?为什么睡一觉就好了? |
| | Could it be a lack of sleep? I felt OK after a good night of sleep, why? |
| Doctor | 颈椎不好会压迫大脑的血管 |
| | A bad cervical spine can compress the blood vessels of the brain. |
| Doctor | 饭量如何?有没有明显变化? |
| | How is your appetite? Is there any noticeable change? |
| Patient | 就是没什么口味 |
| | I just have no appetite. |
| Doctor | 查过胃镜没有? |
| | Do you have taken a gastroscope? |
| Patient | 以前检查有说慢性胃炎 |
| | Previous examinations suggested chronic gastritis. |
| Doctor | 体温多少? |
| | What is the body temperature? |
| Patient | 37度 |
| | 37 degree. |
| Doctor | 精神状态怎样? |
| | How is your mental state? |
| Patient | 还行 |
| | Not bad. |

Table 9: Sequential QA Flow examples in MedQA

| Role | Utterance |
|---|---|
| Patient | 这是什么原因? |
| | What is the reason for the symptom? |
| Doctor | 哪些部位有? |
| | Which parts of your body have the symptom? |
| Patient | 手,和脖子 |
| | Hands and neck |
| Doctor | 局部用炉甘石洗剂外涂,观察下 |
| | Topically apply calamine lotion, and observe it. |
| Doctor | 以前有什么基础疾病吗? |
| | Do you have any previous underlying diseases? |
| Patient | 医生,什么是基础疾病? |
| | Doctor, what is the underlying disease? |
| Doctor | 比如有没有脑梗塞,慢性支气管炎等 |
| | Cerebral infarction, chronic bronchitis, etc. |
| Patient | 没有的,医生。 |
| | No, doctor. |
| Patient | 女,九岁。为什么我的记忆力突然之间变的很差很差了呢? |
| | I'm a nine-year-old female. Why is my memory getting worse suddenly? |
| Doctor | 这样情况多长时间了? |
| | How long has it been like this? |
| Patient | 一个星期了 |
| | It has been a week. |
| Doctor | 问题不大,不用过于紧张担心 |
| | It's not a big problem, don't worry too much. |

Table 10: Clarification/Confirmation examples in MedQA

| Role | Utterance |
|---|---|
| Doctor | 胃痛多长时间了? |
| | How long have you had a stomach ache? |
| Doctor | 是一直痛还是一阵一阵疼痛? |
| | Is it constant pain or bouts of pain? |
| Patient | 二,三天 |
| | two or three days |
| Patient | 疼痛起来特别难受 |
| | Very uncomfortable pain. |
| | |
| Patient | 凝血需要查吗? |
| | Does blood coagulation need to be checked? |
| Patient | 我这些单子里有凝血的检查吗? |
| | Does it contain a coagulation test on these reports of mine? |
| Doctor | 不需要查的 |
| | No need to check |
| Doctor | 没有 |
| | No. |
| | |
| Doctor | 平时以前腰疼吗? |
| | Have you ever had a backache before? |
| Doctor | 腰部疼痛有无牵连到腿疼? |
| | Is the back pain related to the leg pain? |
| Patient | 不疼,就是韧带拉伤以后,腰和腰俩侧疼 |
| | It doesn't hurt. Just after the ligament is strained, the waist and both sides hurt. |
| Patient | 腿不疼 |
| | No leg pain. |

Table 11: Elaboration/Detailing examples in MedQA