# OpenReview forum: "DialogQAE: N-to-N Question Answer Pair Extraction from Customer Service Chatlog"
_EMNLP/2023/Conference — EMNLP 2023 Findings_

### Official Review · Reviewer_psJC · 2023-08-02

**Soundness:** 3

**Excitement:**

2: Mediocre: This paper makes marginal contributions (vs non-contemporaneous work), so I would rather not see it in the conference.

**Paper Topic And Main Contributions:**

This paper primarily introduces the N-to-N question answering pair extraction approach and proposes the DialogQAE model to address this issue. The contribution is that the paper setups a benchmark for DialogQAE with end-to-end and two-stage baselines that support N-to-N QA extraction, as well as the utterance and session-level evaluation metrics.

**Reasons To Accept:**

The dataset in this paper would be beneficial for researchers interested in conducting research on this particular problem.

**Reasons To Reject:**

1.The N-to-N question answer pair extraction approach proposed in this paper is relatively limited.
2.The experimental results show no significant improvement, and the paper lacks a comparison with other methods.
3.This paper is not very clear to understand.

**Reproducibility:**

3: Could reproduce the results with some difficulty. The settings of parameters are underspecified or subjectively determined; the training/evaluation data are not widely available.

**Reviewer Confidence:**

3: Pretty sure, but there's a chance I missed something. Although I have a good feel for this area in general, I did not carefully check the paper's details, e.g., the math, experimental design, or novelty.

---

> ### Author Rebuttal · Authors · 2023-08-29
>
> Thank you for the detailed review. We would like to clarify that:
>
> 1. About N-to-N QA Extraction
>
> We try to unify the question-answer pair extraction approaches, as previous work only focused on limited scenarios, i.e., 1-1 or 1-N, and N-to-N extraction has yet to be explored. Thus, one method may not be applied in other settings, and different settings would require different specific methods. As our method purely leverages the power of T5 model, training for 1-1, 1-N or N-N QA extraction involves no special modifications.
>
> 2. About experimental results and baselines
>
> Our proposed unified QA Extraction is first-of-its-kind. It largely accelerates knowledge accumulation and updates in customer service applications, which has shown great impacts on real-world applications, including education and car sales.
>
> As stated above, existing baselines in the prior work, which are also suitable for both 1-N and N-N scenarios, are unavailable. Therefore, our work provides diverse baselines, consisting of several variations and evaluates a wide range of encoder-only and encode-decoder models with different parameter sizes. We hope it can benefit future studies.
>
> 3. About readability
>
> The structure of the paper is as follows:
>
> 1. Introduction: Provides an overview of the task of N-to-N question and answer (QA) pair extraction from customer service chatlogs. It discusses the importance of QA extraction for enriching knowledge bases and improving dialogue systems. It also highlights the challenges and contributions of the paper.
> 2. Task Overview: Describes the task of N-to-N QA extraction and introduces the concept of exclusive dialogue utterances and speaker role consistency. It categorizes the task into four types based on the number of utterances involved in the extracted QA pairs.
> 3. Methodology: Discusses the tagging-based methods for QA extraction. It explains the end-to-end and two-stage approaches and how they can be used for QA extraction. It also describes the input format and model architectures used in the experiments.
> 4. Experiments: Provides details about the datasets used in the experiments and the evaluation metrics. It presents the experimental results for different models and discusses the performance of the models on different datasets.
> 5. Analysis and Discussions: Analyzes the extracted QA pairs and discusses the dialogue structure and relations between QA pairs. It also explores the domain and language adaptation of the models.
> 6. Related Work: Discusses related work in the areas of QA extraction and dialogue analysis, including research on text-based QA extraction, dialogue information extraction, and dialogue structure analysis.
>
> We would revise our paper and clarify anything that is challenging to follow.

---

### Official Review · Reviewer_AycG · 2023-08-04

**Typos Grammar Style And Presentation Improvements:** In figure 1 I think the symbol in A3 …
**Soundness:** 3

**Excitement:**

3: Ambivalent: It has merits (e.g., it reports state-of-the-art results, the idea is nice), but there are key weaknesses (e.g., it describes incremental work), and it can significantly benefit from another round of revision. However, I won't object to accepting it if my co-reviewers champion it.

**Paper Topic And Main Contributions:**

This paper addresses the problem of acquiring question-answering dialog data from logs of customer service chatbots. The
authors propose to extend existing 1-1 question-answer pair extraction to a more general N-N setting. In the abstract,
in two-party dialogue this involves going from a sequence of <utterance, role> pairs, to a set of <question, answer>
pairs where each member of the pair is a sequence of <utterance, role> pairs.

The paper experiments with two sequence labeling approaches to this task, one which operates end-to-end to label
utterances as questions (Q), answers (A), or neither (O), and the second approach which does this in two stages, the
first labeling as Q or O, and the second labeling the remaining utterances as A or O.

In both cases, a subsequent deterministic procedure groups the labeled utterances into larger <question, answer> clusters based on sequencing and contiguity.

The sequence labeling steps are carried out by various transformer-based systems, and the input is an entire
conversational session.

In general, a large T5 based model was the strongest performer on the two datasets (of five real sets of commercially
collected data) they give detailed results for, on both the single and two-stage versions. Inspection of the results
showed that the extracted QA pairs exhibit different kinds of internal structure familiar from linguistic analysis of
dialogues: clarification, barge-in, etc.

The paper concludes with a brief demonstration that the data acquired in this way improves QA performance on their datasets, and on others.


**Reasons To Accept:**

Acquiring extra training data must be a common problem in the commercial chatbot world, and this paper provides a possibly useful method for acquiring it.

The paper is clearly written and easy to follow

**Reasons To Reject:**

The paper does not offer anything very novel in terms of techniques, but it is nevertheless a useful contribution. I can't think of any strong reason why the paper should be rejected.

**Reproducibility:**

2: Would be hard pressed to reproduce the results. The contribution depends on data that are simply not available outside the author's institution or consortium; not enough details are provided.

**Reviewer Confidence:**

3: Pretty sure, but there's a chance I missed something. Although I have a good feel for this area in general, I did not carefully check the paper's details, e.g., the math, experimental design, or novelty.

---

> ### Author Rebuttal · Authors · 2023-08-29
>
> Thank you for your detailed review. We would like to emphasize that:
>
> 1. The technical contribution of our work
>
> We leverage the feature of T5, span recovery, to address the task of QA extraction. Our method involves no extra modification of the original language model, making it easy to implement. It also echoes the current trend of converting most, if not all, Natural Language Processing tasks into Language Modeling. The success of our application stresses the importance and necessity of adding a span corruption / fill-in-the-blank task during pre-training, besides traditional pure left-to-right language modeling task, for better support of sequence-tagging NLU tasks. We will take your suggestion and fix the error in Figure 1.
>
> We also would like to highlight that other main contributions include:
>
> - We set up a benchmark for DialogQAE with diverse baselines.
> - We show that DialogQAE is an effective paradigm for dialogue analysis.
> - We find that the proposed DialogQAE model effectively automates and accelerates the cold-start or upgrade of a commercial dialogue system.
>
> 2. Regarding to the **Reproducibility of our work,** we would open-source the code, model and public-available dataset.

---

### Official Review · Reviewer_VBi6 · 2023-08-04

**Soundness:** 4

**Excitement:**

4: Strong: This paper deepens the understanding of some phenomenon or lowers the barriers to an existing research direction.

**Paper Topic And Main Contributions:**

The paper proposes an approach to deal with N-to-N QA extraction tasks in which the derived questions and corresponding answers might be separated across different utterances. The paper also introduces a suite of generative and discriminative tagging methods with end-to-end and two-stage architecture. The suitе is tested on 5 customer service datasets and set up a benchmark for N-to-N DialogQAE with utterance and session-level evaluation metrics. Analysis of extracted QA pairs shows that the relations between and inside the QA pairs can be indicators to analyze the dialogue structure, e.g. information seeking, clarification, barge-in, and elaboration. The proposed models are tested on dialogs from different domains and languages.

**Reasons To Accept:**

•	A well-structured and well-written paper.
•	The topic is of research and applied interest.
•	The experiments are well-designed and the results are analyzed.
•	A suite is proposed that set up a benchmark for N-to-N DialogQAE with utterance and session-level evaluation metrics
•	The proposed models can adapt to different domains and languages, and reduce the labor cost of knowledge accumulation in the real-world product dialogue platform.


**Reasons To Reject:**

-

**Reproducibility:**

4: Could mostly reproduce the results, but there may be some variation because of sample variance or minor variations in their interpretation of the protocol or method.

**Reviewer Confidence:**

3: Pretty sure, but there's a chance I missed something. Although I have a good feel for this area in general, I did not carefully check the paper's details, e.g., the math, experimental design, or novelty.

---

> ### Author Rebuttal · Authors · 2023-08-29
>
> Thank you for the thoughtful and constructive feedback on our submission.

---

### Meta-Review · Area_Chair_9fgX · 2023-09-12

**Recommendation:** 3

**Metareview:**

This paper tackles the challenge of extracting question-answering dialogue data from customer service chatbot logs. The authors propose an extension of the existing 1-1 question-answer pair extraction to a more versatile N-N setting. Furthermore, the paper introduces a suite of generative and discriminative tagging methods, employing both end-to-end and two-stage architectures. These methods are evaluated on five customer service datasets. An in-depth analysis of the extracted QA pairs reveals valuable insights into the dialogue structure, including information-seeking, clarification, barge-in, and elaboration. The proposed models are subjected to testing on dialogues from diverse domains and languages. While reviewers acknowledged that the paper does not present groundbreaking techniques, it undeniably constitutes a valuable contribution to the field.

---

### Decision · Program_Chairs · 2023-10-07

**Decision:**

Accept-Findings

**Comment:**

This paper tackles the challenge of extracting question-answering dialogue data from customer service chatbot logs. The authors propose an extension of the existing 1-1 question-answer pair extraction to a more versatile N-N setting. Furthermore, the paper introduces a suite of generative and discriminative tagging methods, employing both end-to-end and two-stage architectures. These methods are evaluated on five customer service datasets. An in-depth analysis of the extracted QA pairs reveals valuable insights into the dialogue structure, including information-seeking, clarification, barge-in, and elaboration. The proposed models are subjected to testing on dialogues from diverse domains and languages. While reviewers acknowledged that the paper does not present groundbreaking techniques, it undeniably constitutes a valuable contribution to the field.